# Low Levels of Urinary PSA Better Identify Prostate Cancer Patients

**DOI:** 10.3390/cancers13143570

**Published:** 2021-07-16

**Authors:** Sergio Occhipinti, Giulio Mengozzi, Marco Oderda, Andrea Zitella, Luca Molinaro, Francesco Novelli, Mirella Giovarelli, Paolo Gontero

**Affiliations:** 1Department of Molecular Biotechnologies and Health Sciences, University of Turin, 10126 Turin, Italy; franco.novelli@unito.it (F.N.); mirella.giovarelli@unito.it (M.G.); 2NIB Biotec srl, 10135 Turin, Italy; 3Clinical Biochemistry Laboratory, Department of Laboratory Medicine, AOU Città della Salute e della Scienza di Torino, 10126 Turin, Italy; gmengozzi@cittadellasalute.to.it; 4Division of Urology, Department of Surgical Science, AOU Città della Salute e della Scienza di Torino, University of Turin, 10126 Turin, Italy; marco.oderda@unito.it (M.O.); azitella@cittadellasalute.it (A.Z.); paolo.gontero@unito.it (P.G.); 5Division of Pathology, AOU Città della Salute e della Scienza di Torino, 10126 Turin, Italy; luca.molinaro@unito.it

**Keywords:** prostate cancer prevention, prostate cancer detection, screening, biomarkers, diagnosis, early detection

## Abstract

**Simple Summary:**

Elevated PSA levels in blood tests are the gold standard for early prostate cancer detection, but its lack of specificity limits its clinical use as a mass screening test. The paradox is that it has long been known that advanced prostate cancers can lose PSA expression. We have observed that in the presence of tumors, the prostate produces and secretes less PSA than in healthy or benign conditions. Therefore, the PSA evaluation in urine provided more accurate information on the presence of prostate tumors than the blood test, representing a new method for the screening of prostate cancer.

**Abstract:**

Serum prostatic specific antigen (PSA) has proven to have limited accuracy in early diagnosis and in making clinical decisions about different therapies for prostate cancer (PCa). This is partially due to the fact that an increase in PSA in the blood is due to the compromised architecture of the prostate, which is only observed in advanced cancer. On the contrary, PSA observed in the urine (uPSA) reflects the quantity produced by the prostate, and therefore can give more information about the presence of disease. We enrolled 574 men scheduled for prostate biopsy at the urology clinic, and levels of uPSA were evaluated. uPSA levels resulted lower among subjects with PCa when compared to patients with negative biopsies. An indirect correlation was observed between uPSA amount and the stage of disease. Loss of expression of PSA appears as a characteristic of prostate cancer development and its evaluation in urine represents an interesting approach for the early detection of the disease and the stratification of patients.

## 1. Introduction

Prostate cancer (PCa) represents the second most common malignancy for men worldwide with 375,304 death in 2020 [1]. Early diagnosed localized disease can be successfully cured by radical surgery or radiation; however, the majority of locally advanced and all metastatic diseases are treated with androgen deprivation. PCa mortality has decreased in the past years, mainly due to the widespread use of preliminary exams [2]. However, balancing the early detection of potentially lethal PCa that may benefit from therapy with low-risk cancers that suffer complications from unnecessary treatment continues to be the controversy regarding PCa screening. The decision to undergo testing to detect PCa is based on preliminary exams, such as digital rectal examination, or, more typically, an elevated serum PSA. However, PSA is an organ- but not cancer-specific biomarker, and its serum levels can also be affected by non-malignant prostatic pathologies. In this regard, the use of PSA screening for PCa has declined recently because of concerns about over-diagnosis and over-treatment [3,4,5].

In the last decade, various serum and urine biomarkers were studied to aid in assessing the risk of harboring a clinical significant prostate cancer. Several novel tests became commercially available, but none of these were routinely used due to limited evidences about benefits over the standard of care on the general population [6].

Recent studies highlighted that infertility in men could be related to the risk to develop high aggressive prostate cancer [7]. Prostate is a gland that produces seminal fluid content, fundamental for the survival, motility and quality of spermatozoa. It is already known that chronic inflammation affects chemical features of the liquid part of the ejaculate, affecting fertility [8]. However, the correlation between infertility and prostate cancer remains unclear.

During neoplastic transformation, cells undergo deep changes both phenotypic and functional. The majority of prostatic neoplasia are adenocarcinoma, that is a tumor involving changes of glandular cells. This evidence suggests that neoplastic transformation in the prostate can compromise the composition of prostatic fluid.

It has long been known that high-grade prostate cancers can lose PSA expression [9,10] and more detailed studies using the tissue microarray technology have confirmed that the expression of the PSA in the tissue can be used both as a diagnostic and prognostic parameter for prostate cancer [11].

This fact results in the paradox of a high serum PSA value and the loss of PSA expression in the cancer tissue being regarded as a poor prognostic sign. This discrepancy might explain some of the difficulties with the diagnostic usefulness of serum PSA measurement alone.

Since the urinary tract is in close contact with the prostate, the factors produced by the prostatic tissue can be transferred and therefore detected in the urine, representing biological markers for the diagnosis and prognosis of prostate cancer.

On the basis of this, measurement of the urinary PSA should provide useful information concerning the physiology and pathological situation of the prostate, since the PSA is the normal product of the epithelial cells that surround the prostatic acini and ducts [12]. In a work published in 2007, it was observed that the ratio between urinary PSA and plasmatic PSA is different between patients with prostate tumor, patients with benign hyperplasia of the organ and healthy subjects [13]. However, the absence of prostate massage prior to the collection of urine does not allow a consistent prostatic secretion to be obtained.

In the present study, urine samples were collected from consecutive men scheduled for prostatic biopsy based routine parameters, as elevated total PSA level or abnormal digital rectal examination or suspect multiparametric magnetic resonance. Urinary PSA (uPSA) were quantified in order to evaluate a correlation with the presence of prostate cancer.

The mean levels of uPSA were lower in patients compared to healthy individuals, with a negative trend for the increase in tumor stage.

These results sustain the idea that the analysis of PSA in prostatic secretion instead of peripheral blood could be an interesting method for the screening of prostate cancer and rethink the concept of the less effective PSA test.

## 2. Materials and Methods

### 2.1. Study Population and Study Design

Subjects were enrolled from the urology clinic in the AOU Città della Salute e della Scienza di Torino Hospital, Turin, Piedmont, Italy. Men who were scheduled for prostate biopsy from 1 May 2015 to 30 April 2018 (3 years) were invited to participate in the study. Biopsy indication was decided according to clinical practice, including information on PSA levels, digital rectal examination (DRE) findings, prostate volume, family history and MRI imaging. Urine samples were collected after a standardized DRE and before the biopsy. Histological specimens consisted of 10–12 core biopsy samples obtained with ultrasound guidance. Participants (*n* = 574) were divided in two cohorts on the basis of the year of enrollment: 376 men in cohort A (training) and 198 in cohort B (validation).

### 2.2. Sample Collection and Processing

First, 30 mL of voided urine were collected after prostate massage to extract prostatic secretions, through three digital compressions in each lobe starting from the base, moving downwards to the middle and the apex in a timelapse of 30 s.

After a gentle shake of the sample, an aliquot of 15 mL was taken and stored in Falcon tubes at −80 °C within 5 min from collection.

### 2.3. Urine Analysis

Urinary measurements of PSA were performed by means of ELISA assay (R&D Systems, R&D Systems, Inc., Minneapolis, MN, USA) after optimization on urine matrix following the manufacturer’s instruction, and confirmed by an electrochemiluminescence immunoassay (ECLIA) employing a Cobas^®^ laboratory-platform (Roche Diagnostic GmbH, Mannheim, Germany) in the Biochemical and Clinical Laboratory (Baldi and Riberi), AOU Città della Salute e della Scienza di Torino, Turin, Italy.

### 2.4. Immunohistochemistry

A series of 10 prostate carcinoma cases from surgical resected prostate was collected between 2015 and 2017 from the files of the Pathology Institute at the Department of Medical Sciences of the University of Torino. Cases were selected between 3 + 4 and 4 + 5 Gleason score. All cases were first surgical diagnoses, without previous neaodiuvant therapy. Prostate carcinomas were classified according to the WHO Classification of Tumours of the Urinary System and Male Genital Organs 4th Edition 2016 criteria. Specimens had been fixed in 4% buffered formaldehyde, routinely processed, and embedded in paraffin. Three-micron sections were collected on Superfrost plus slides and used for immunohistochemical analysis. PSA immunoreactivity was studied in all cases, and the immunohistochemical reaction, using antibody anti-PSA (ER-PR8 clone, mouse monoclonal antibody Cell Marque-Roche), was performed in an automated immunostainer (Ventana BenchMark Auto-Stainer, Ventana Medical Systems, Tucson, AZ, USA). PSA staining was scored as positive or negative, and in case of positivity, a three-tiered positivity score was performed based on cell percentage positivity; normal prostate glands within the specimens were used as internal positive controls.

### 2.5. Ethics Statement

The human studies were conducted according to the Declaration of Helsinki principles. Human investigations were performed after approval of the study by the Scientific Ethics Committee of A.O.U. Città della Salute e della Scienza di Torino, A.O. Mauriziano, A.S.L. TO1 (Prot. No. 0110644). Written informed consent was received from each participant before inclusion in the study and specimens were anonymized before analysis.

### 2.6. Study Endpoints and Statistical Analyses

Categorical variables were described using frequencies and proportions, while continuous values were expressed using mean, median and interquartile range (IQR). We categorized subjects by category risk based on PSA value, Gleason Score (GS), number of positive biopsies, and TNM staging [14]. To test whether urine PSA levels were different between histologically negative subjects and histologically positive subjects (stratified by category risk), we performed the Kruskal–Wallis and Dunn tests on the log-transformed PSA levels. Subjects from the two cohorts were analyzed separately.

Then, we performed multivariate logistic regression analysis to evaluate the urine PSA diagnostic performance of detecting clinically significant PCa (GS ≥ 7 and/or Spsa > 10 ng/mL). The discriminative power of the logistic model was assessed by calculating the area under the receiver operating characteristic (ROC) curves (AUC). We compared the diagnostic performance of three different multivariate logistic regression models including known risk factors of prostate cancer. The first model (SOC model) included serum PSA levels, age at diagnosis and abnormal DRE. The second model (uPSA model) included urine PSA (uPSA) levels. The third model (uPSA + SOC model) included both serum PSA and uPSA levels as well as age and abnormal DRE. Comparisons of AUCs provided by different models were determined using DeLong’s method. Logistic regression coefficients were estimated in Cohort A, while Cohort B was used for external validation.

In order to estimate potential optimism introduced by overfitting, the predictive models were internally validated through bootstrap method (1000 bootstrap samples).

Statistical analyses were performed with MedCalc^®^ Statistical Software version 19.8 (MedCalc Software Ltd., Ostend, Belgium).

## 3. Results

### 3.1. Patient Characteristics

A total of 574 men scheduled for prostate biopsy were enrolled, but only subjects with PSA levels below 25 ng/mL (*n* = 527) were included in the analysis. Men were divided into training Cohort A (*n* = 348) and validation Cohort B (*n* = 179). For Cohort A, of 348 men, 175 men had negative biopsy (95 no evidence of tumor or benign prostatic hyperplasia, 42 inflammation/prostatitis, 21 high grade prostatic intraepithelial neoplasia and 17 Atypical small acinar proliferation) and 173 men (49.7%) had a positive biopsy outcome.

According to PSA, GS and tumor staging [14], PCa patients were stratified into low risk (*n* = 20), favorable-intermediate (fav-int) risk (*n*=53), unfavorable-intermediate (un-int) risk (*n* = 47), and high risk PCa (*n* = 53). The prevalence of clinically significant (cs) PCa was 44% (Table 1).

For Cohort B, of 179 men, 79 men had negative biopsy (31 no evidence of tumor or benign prostatic hyperplesia, 30 inflammation/prostatis, 11 high grade prostatic intraepithelial neoplasia and 7 Atypical small acinar proliferation) and 100 men (55.9%) had a positive biopsy outcome, of which 15 low risk, 32 fav-int risk, 30 un-int risk, 23 high risk PCa. The prevalence of csPCa was 47.5% (Table 1).

In Cohort A e Cohort B recruited subjects had an average age of 68 years and a similar DRE positivity (39.7% vs. 37.4%) and prevalence of csPCa (44% vs. 47.5%).

The mean PSA value was 7.0 ng/mL and 8.1 ng/mL in Cohort A and Cohort B, respectively, with a slight difference between the two groups (*p* = 0.04).

### 3.2. Correlation Between Urinary PSA and PSA Expression in Prostate Tissue

Immunohistochemistry (IHC) analysis was performed to investigate the expression of PSA protein in prostatic tissue after prostatectomy.

All the samples probed with PSA antibody showed positive staining with different immunoreactivity intensity classified as 1+, 2+ or 3+ staining.

We observed a positive correlation (*r* = 0.67, *p* = 0.03) between PSA staining intensity in prostate tissue and PSA amount measured in urine (Figure 1A).

As shown in Figure 1B, low and high intensity of PSA in tissue reflect the low and high amount of PSA in urine, respectively.

### 3.3. Differences in Urinary PSA between Healthy Individuals and Patients with Advanced Prostate Cancer

We already preliminary observed that the amount of uPSA could predict repeat prostate biopsy outcome [15].

In order to prove if a decrease in uPSA levels could be a sign of the development of tumors in the prostatic gland, we analyzed urine samples from patients with advanced prostate tumors (Gleason Score ≥ 8, and/or presence of metastases, and/or T3 stage) in comparison to urine collected from subjects with at least 2 prostatic biopsies without presence of cancer.

As shown in Figure 2A, the mean levels of uPSA were significantly lower (*p* = 0.0002) in patients with advanced prostate cancer (advPCa) compared to cancer-free subjects (Healthy). Next, we used a Receiver-operating characteristic (ROC) analysis to assess the diagnostic capability of uPSA in PCa. The analysis showed the area under the ROC curve (AUC) for uPSA was 0.786 (Figure 2B).

Similarly we evaluated the amount of serum PSA. The mean levels were significantly higher in advPCa compared to Healthy (*p* = 0.007, Figure 2C), and the analysis showed the area under the ROC curve (AUC) for uPSA was 0.713 (Figure 2D).

### 3.4. Quantification of Urinary PSA in Subjects Candidate for Prostate Biopsy

Urinary samples from men suspected of prostate cancer were collected before prostate biopsy and tested for the presence of PSA. We observed a negative trend in uPSA levels for the increase in class risk (*p* for trend 0.0027, Figure 3, Table 2).

The mean levels of uPSA were significantly decreased in patients with fav-int, un-int and high risk PCa compared to healthy subjects, with *p* equal to 0.0111, 0.0018, 0.0001, respectively, and compared to low risk PCa, with *p* equal to 0.0119, 0.0035, 0.0003, respectively, whilst no difference were observed between low risk PCa and control group (Figure 3, Table 2).

### 3.5. Evaluation of Urinary PSA, Serum PSA, Age and DRE in Subjects Candidate for Prostate Biopsy

We evaluated the levels of uPSA in patients with clinically significant PCa (csPCa) and subjects without sign of PCa or low risk PCa (non-cancer).

uPSA levels were significantly lower in csPCa patients than in non-cancer individuals (Figure 4A; *p* = 0.0001).

Standard clinical parameters as serum PSA (Figure 4B), age (Figure 4C) and positive DRE rate (Figure 4D) were significantly higher in csPCa patients compared to non-cancer, with a *p* value of 0.028, 0.029 and 0.007, respectively.

Pearson’s correlation analysis was performed to determine whether there was correlation between the levels of serum PSA and age and uPSA. No significant correlation was found between uPSA and serum PSA or age (Figure 5).

These results suggest that uPSA could be an indicator of PCa progression and combined with routine parameters such as serum PSA.

### 3.6. Diagnostic Accuracy of Urinary PSA Levels in Patients with Clinically Significant Prostate Cancer

We assessed the role of uPSA as a biomarker for discrimination between csPCa and non-cancer subjects.

We used a receiver-operating characteristic (ROC) analysis to assess the diagnostic capability of uPSA in csPCa. The analysis showed the area under the ROC curve (AUC) for standard of care parameters (as PSA, DRE and age, SOC) and uPSA alone were 0.612 and 0.691, respectively.

Then, uPSA in comparison and in conjunction with SOC (uPSA+ SOC) were further analyzed for csPCa detection. As shown in Table 3 and Figure 6, the AUC of both uPSA alone and uPSA + SOC (0.721) was significantly higher than the AUC of SOC.

Internal validation, using the bootstrap method with 1000 resamples, for the different models showed an optimism estimated 0.001 (Table 3).

Taken together, these results suggested that uPSA performs better than SOC in the detection of clinical significant PCa and can improve the diagnostic capability of routine parameters.

Positive predictive value (PPV) and negative predictive value (NPV) were calculated to identify the proportion of patients correctly predicted to have csPCa and the proportion of patients correctly predicted to be free from disease or have an indolent PCa, respectively.

Considering a fixed sensitivity of 95% or 90%, the specificity for uPSA and uPSA + SOC were greater than SOC alone.

Table 4 lists the PPV and NPV at different cut-offs for SOC, uPSA and uPSA + SOC, showing a better accuracy of uPSA and the combination in correctly identifying biopsies outcome.

A new statistic, termed the “number needed to predict” (NNP), represents the number of patients who need to be examined in a defined population in order to correctly predict the diagnosis of one person. NNP is dependent on prevalence and may therefore be deemed a better descriptor of diagnostic tests in patient populations with different prevalence of disease [16].

At a cut-off of 95% and 90% of sensitivity the NNP was higher for SOC compared to uPSA or uPSA + SOC (Table 4).

This evidence suggests that a lower number of men should be examined by further clinical exams (e.g., prostate biopsy) to correctly identify csPCa patients by evaluating uPSA alone or in combination with routine parameters.

Although no correlations were observed between urinary and serum PSA, we evaluated if uPSA had different diagnostic capability at different cut-off of sPSA. Recruited subjects were divided into three groups: 51 with serum PSA level ≤ 4 ng/mL, 238 with serum PSA level > 4.1 ng/mL and <10 ng/mL, 59 with serum PSA level >10.1 ng/mL and <25 ng/mL. The urinary PSA levels were compared between men with non-cancer and patients with csPCa. ROC analysis for uPSA in men with sPSA between 0–4 ng/mL, 4.1–10 ng/mL and 10.1–25 ng/mL were conducted. The AUCs were 0.602, 0.683 and 0.792, respectively (Table 5).

### 3.7. Development and Validation of a Diagnostic Model in Subjects Candidate for Prostate Biopsy

Based on the four independent variables, three diagnostic models (SOC, uPSA, Upsa + SOC) were established for the risk assessment of csPCa in men candidates for prostate biopsy.

The performance characteristics of these models were evaluated in an independent validation Cohort B. The SOC, based on the model incorporating PSA, age and DRE was used as the main reference. The AUC for SOC in predicting the chance of csPCa was 0.598 (Table 6). The AUC for uPSA was 0.720 and the AUC of uPSA + SOC was 0.666, which were significantly higher than SOC alone with *p* = 0.025 and *p* = 0.002, respectively (Table 6 and Figure 7).

### 3.8. Clinical Utility of Urinary PSA

As observed in the training cohort, uPSA and the combination of uPSA-SOC show better PPV and NPV compared to SOC (Table 7).

Moreover, at a cut-off of 95% and 90% of sensitivity the NNP was 3 times lower for uPSA or uPSA + SOC than SOC (Table 7).

These evidence enforce the capability of uPSA to correctly identify csPCa patients better than routine parameters.

To evaluate the clinical utility of uPSA different cut-off were considered. At a cut-off of 40% of probability for csPCa, a total reduction of unnecessary biopsies by 27% were obtained without missing any cancer (Table 8).

### 3.9. Combination of uPSA with Multiparametric Magnetic Resonance Results

We assessed the possibility to combine uPSA analysis with MRI results and conventional routine parameters. In the subgroup that underwent prostate biopsy after MRI (*n* = 184) 80 were free from disease, 13 had low risk PCa, 37 Int-fav PCa, 34 Int-unfav PCa, 20 high risk PCa.

On those subjects, 65% of patients with PiRADS-5, 50% of patients with PiRADS-4 and 24% of patients with PiRADS-3 had csPCa (Figure 8A).

The MRI results in conjunction with standard clinical parameters as PSA, DRE and age (SOC) displayed similar diagnostic performance in detect csPCa compared to MRI alone (Figure 8B; Table 9). The combination of MRI with SOC and uPSA analysis demonstrated higher diagnostic performance with an AUC of 0.698 that was significantly higher than SOC, MRI and the combination of both alone (Table 9).

### 3.10. Diagnostic Accuracy of Urinary PSA Levels in Men Undergoing Repeat Prostate Biopsy

In order to evaluate the clinical utility of uPSA in reduce unnecessary repeat biopsy uPSA model were applied on patients with a prior negative biopsy underwent to a second prostate biopsy. Of 56 patients 27 had no evidences of PCa, 8 received a diagnosis of low risk tumors, 10 int-fav, 7 int-sfav, and 4 high risk.

At a cut-off of 40% and 50% of probability for csPCa, a potential reduction of unnecessary biopsies by 43% and 71%, respectively, were obtained without missing any high grade cancer (Table 10).

## 4. Discussion

Real benefits of PSA-screening and its impact on mortality are still controversial. The United States Preventive Services Task Force (USPSTF) recommends against its use in the general population [17], the European study demonstrated an impact on reducing mortality [18,19], while the PLCO study [20] and the USPSTF [21] did not demonstrate such benefits.

Although it allows for early diagnosis of advanced cancers, screening also detects indolent cancers that potentially do not require any intervention, with a strong impact on quality of life and in costs for the health system [22,23].

During the last decade, numerous biomarkers have been developed and their use has been assessed all along the disease states. Several serum PSA derivatives have been evaluated in an effort to improve the specificity of PSA. The clinical utility of total PSA and free PSA, proPSA, with its most stable form (−2) proPSA, and intact PSA has been widely tested and used to design of the Prostate Health Index (PHI) [24] and 4-kallikrein (4K) score [25]. In addition to traditional blood, urine has been proposed as a suitable source for prostatic biomarkers. To date, several new promising urinary biomarkers have been identified and considered for early diagnosis of prostate cancer [26].

PCA3 assay was the first molecular urine test approved by FDA to help determine the need for repeat prostate biopsies in men with a previous negative biopsy [27]. Although PCA3 is a reliable tool for detection of PCa, studies have noted that there is no correlation with aggressiveness of the cancer or clinical tumor stage.

More recently other urine-based biomarkers as ExoDx Prostate [28], SelectMDx [29], and the Mi-prostate score [30] has been developed, but the dependence on some routine parameters, such as PSA value, age, prostate volume, makes them additional tests that find applications only in selected cases.

Even if some of them have shown good predictive values, most are not yet validated and scientifically mature for daily practice and therefore not recommended by guidelines.

The measurement of urinary PSA represents an interesting parameter to obtain information about prostate physiology and pathology, as the presence of neoplastic transformation. PSA belongs to kallikreins, molecules produced by epithelial cells lining the acini and prostatic ducts [12]. In pathologic situations, such as chronic inflammation or neoplasia, many events characterized by compression, neovascularization and disruption of the prostatic ducts, the polarity of the epithelial cells should be inverted to release the secreted kallikreins across the basement membrane, thus reaching the bloodstream. However, when the architecture of the gland is not sufficiently corrupted, PSA does not increase in serum.

Several studies have demonstrated that low PSA expression by immunohistochemistry is associated with a greater Gleason score and raised cell proliferation [31,32].

Erbersdobler and colleagues observed that preoperative serum PSA value does not correlate with PSA expression in the tumor specimens. On the contrary, the loss of PSA expression was significantly associated with aggressiveness of tumors and with biochemical recurrence after prostatectomy [32].

In the current study, PSA amounts were evaluated in urine derived from men with a suspicion of prostate cancer after digital rectal exploration.

The level of PSA detectable in urine is directly correlated to PSA expression in prostate tissue and is uncorrelated to PSA levels in blood. This aspect underlines that probably standard evaluation of PSA is less representative of the health state of the prostate compared to urine.

Here, we showed that uPSA is lower in patients with PCa compared to men without signs of cancer, with a gradual decrease among increasing stages of tumors. Importantly, no differences were observed between healthy individuals and patients with low-risk tumors, suggesting the possibility to discriminate clinically relevant from indolent PCa.

Evaluation of uPSA demonstrated a higher diagnostic capability compared to standard parameters, as serum PSA, age and digital-rectal exploration. Moreover, the combination of uPSA with the standard of care gives better diagnostic results. uPSA showed an added value also in conjunction with multiparametric MRI, which has become a key exam in the diagnosis of PCa, with the majority of biopsies being now performed under TRUS and MRI guidance (the so-called “fusion biopsies”). mpMRI has been shown to outperform conventional TRUS biopsy in both biopsy-naïve and biopsy-experienced patients, with a higher detection rate of csPCa and a lower detection of unsignificant cancers [33]. However, mpMRI is not flawless and is not able to diagnose all cancers, even if clinically significant. Theoretically, mpMRI does not detect lesions with low cellular density that should reflect low grade cancers, generally considered clinically unsignificant. Nevertheless, not all low grade cancers are the same and not all of them can be addressed to active surveillance, especially if a high volume of disease is present. In this setting, a biomarker such as uPSA could be helpful to trigger a prostate biopsy, or guide the choice between surveillance and treatment.

The main limitation of this study is the fact that most patients underwent a conventional TRUS-guided biopsy, which is limited by a 20% false negative rate [34], and only a minority of them was diagnosed with a MRI/TRUS fusion biopsy. It would be interesting to test the diagnostic performance of uPSA in a large series of patients that undergo the current diagnostic pathway that includes last-generation multiparametric MRI [35]. Furthermore, even if the current study demonstrated an improved PCa diagnostic ability with the implementation of uPSA, the AUCs remained quite low, witnessing room for improvement.

All these things considered, here we showed for the first time that loss of PSA expression and production by the prostate could be an hallmark of prostate cancer and PSA measurement in urine appears to be a more effective method in the diagnosis or follow-up of prostate cancer compared to evaluation in blood.

## 5. Conclusions

We found an interesting association among the loss of PSA expression in prostate tissue, urinary amount of PSA and the presence and stage of prostate cancer.

In the biopsy scenario and in the treatment decision making of patients, evaluation of uPSA might be a useful parameter for predicting diagnosis and prognosis.

## Figures and Tables

**Figure 1 cancers-13-03570-f001:**
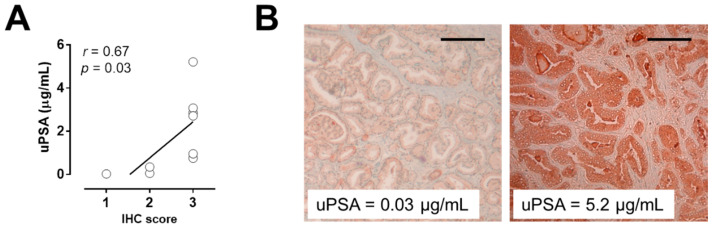
Correlation between PSA expression in prostate tissue and urine: urine PSA (uPSA) amount and PSA expression in prostate tissue derived from the same patient (*n* = 10) (**A**). Two representatives immunohistochemical staining of prostate tissue after radical prostatectomy with low (left) and high (right) expression of PSA (**B**). Magnification 40×. Scale bar = 100 μm.

**Figure 2 cancers-13-03570-f002:**
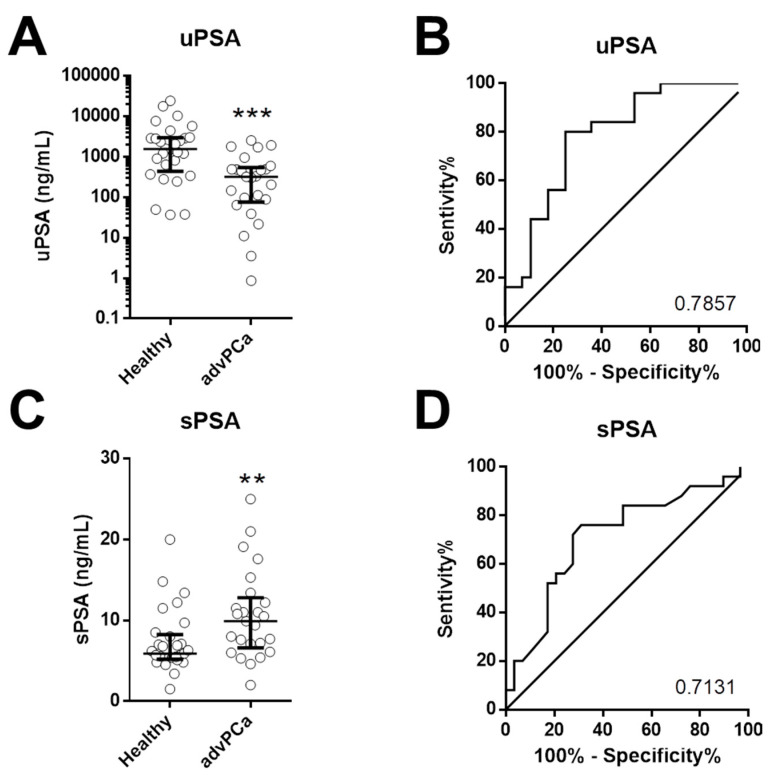
Urinary and serum PSA in healthy individuals and advanced PCa: uPSA in Healthy individuals (*n* = 28) and men with advanced PCa (advPCa, *n* = 21) (**A**). *** *p* < 0.0001. ROC for uPSA in detecting advPCa (**B**). AUC=0.7857. Serum PSA (sPSA) in Healthy individuals (*n* = 28) and men with advanced PCa (advPCa, *n* = 21) (**C**). ** *p* < 0.001. ROC for uPSA in detecting advPCa (**D**). AUC = 0.7131.

**Figure 3 cancers-13-03570-f003:**
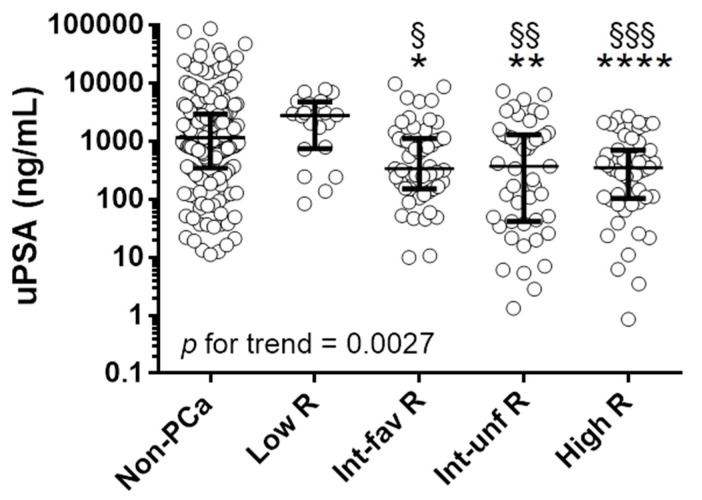
Urinary PSA in healthy individuals and patients with PCa: uPSA in men with no evidence of PCa (Non-PCa, *n* = 175), low- (Low-R, *n* = 20), intermediate-favorable- (Int-fav-R, *n* = 53), intermediate-unfavorable (Int-unf-R, *n* = 47), high- (High-R, *n* = 53) risk patients. * *p* < 0.05, ** *p* < 0.01, **** *p* < 0.0001 to Non PCa. ^§^ *p* < 0.05, ^§§^ *p* < 0.01, ^§§§^ *p* < 0.001 to Low-R. *p* for the trend = 0.0027.

**Figure 4 cancers-13-03570-f004:**
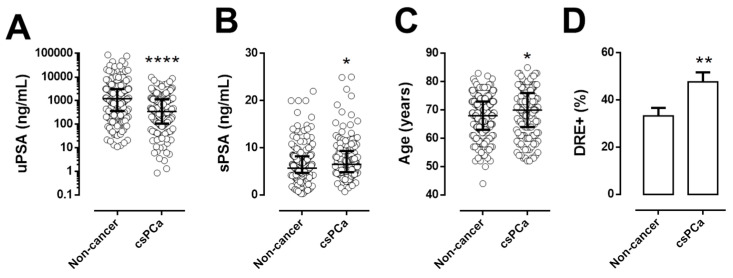
Urinary PSA and standard parameters in patients with clinical significant PCa: amount of uPSA (**A**), serum PSA (sPSA) (**B**), years of age (Age) (**C**), percentage of suspect DRE (DRE+) (**D**) in men without prostate cancer or low-risk PCa (Non cancer, *n* = 195) and patients with clinical significant PCa (csPCa, *n* = 153). * *p* < 0.05, ** *p* < 0.01, **** *p* < 0.0001, to non-cancer.

**Figure 5 cancers-13-03570-f005:**
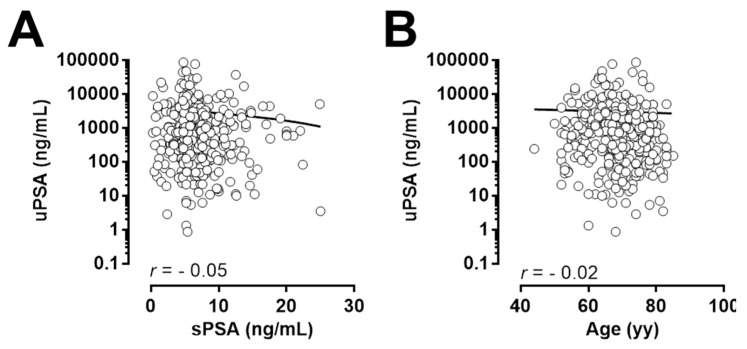
Correlation between uPSA and standard parameters: uPSA and sPSA (**A**) and uPSA and Age (**B**) in subjects candidate for prostate biopsy.

**Figure 6 cancers-13-03570-f006:**
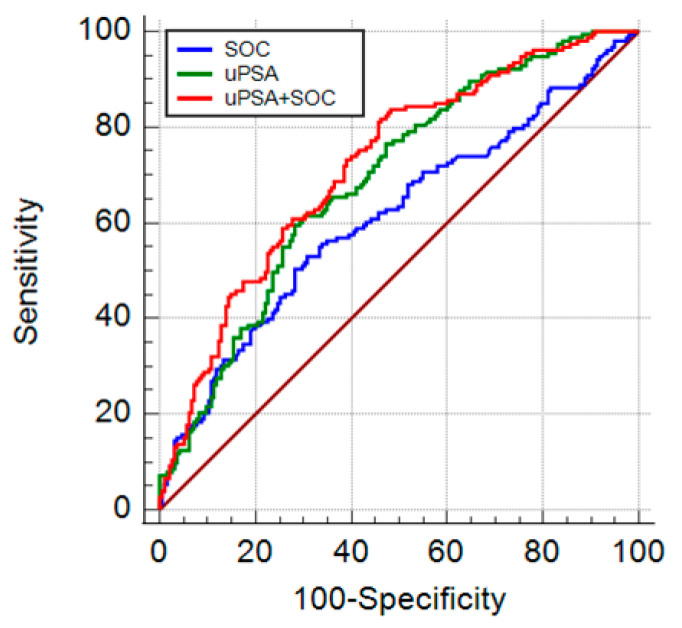
Diagnostic performance in detecting csPCa: AUC ROC for standard parameters SOC (sPSA, age, DRE, blu line), uPSA (green line), combination of uPSA and SOC (uPSA + SOC, red line).

**Figure 7 cancers-13-03570-f007:**
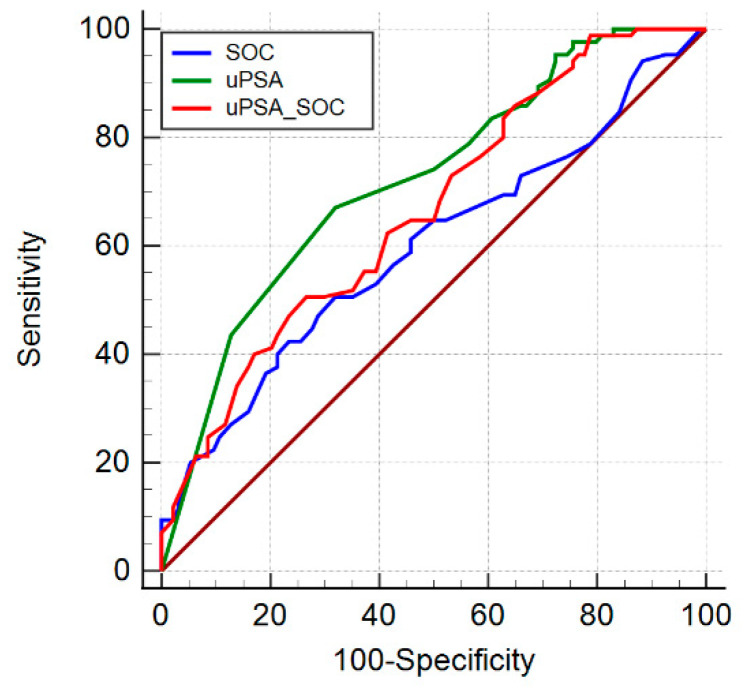
Diagnostic performance in detecting csPCa in validation cohort: AUC ROC for standard parameters SOC (sPSA, age, DRE, blu line), uPSA (green line), combination of uPSA and SOC (uPSA + SOC, red line).

**Figure 8 cancers-13-03570-f008:**
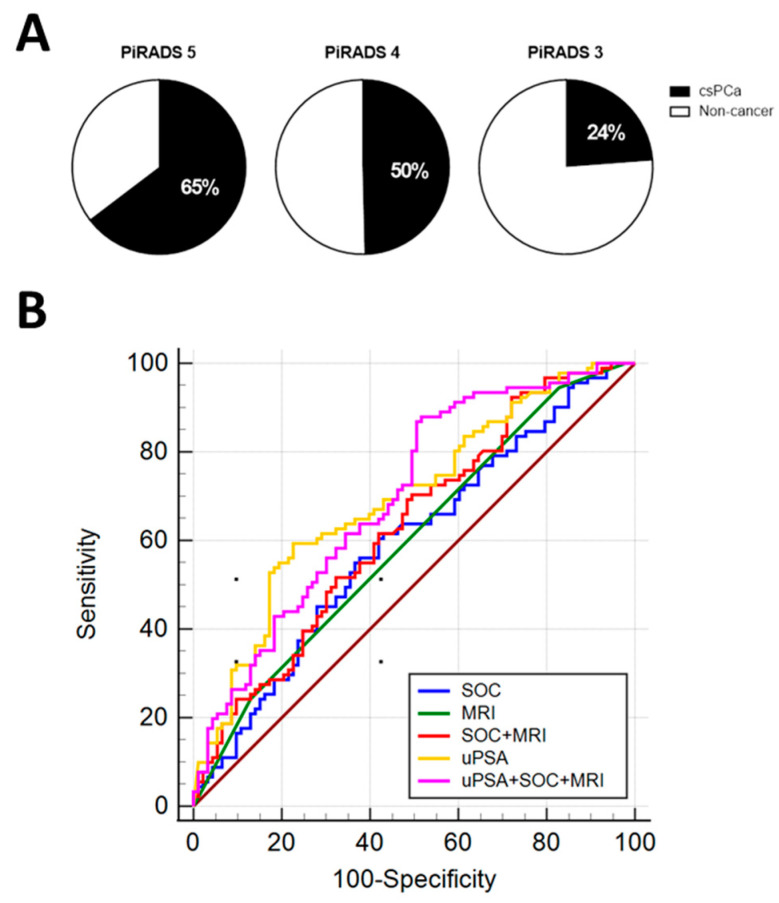
Diagnostic performance of uPSA and MRI: Percentage of csPCa patients with PiRADS 5, 4 and 3 (**A**). AUC ROC for SOC (blu line), magnetic resonance (MRI, green line), combination of SOC and MRI (red line), uPSA (yellow line), combination of uPSA, SOC and MRI (pink line) (**B**).

**Table 1 cancers-13-03570-t001:** Patient characteristics.

Characteristics	Cohort A	Cohort B	*p*
Patients, *n*	376	198	-
Evaluable samples, *n* (%) ^1^	348 (92.6)	179 (90.4)	-
Age, years, mean (median; IQR)	68 (68; 63–74)	68 (69; 63–74)	ns ^2^
PSA, ng/mL, mean (median; IQR)	7 (6; 4.8–8.4)	8.1 (6.8; 4.8–10)	0.04
DRE adnormal, *n* (%)	138 (39.7)	67 (37.4)	ns ^2^
PCa diagnosis, *n* (%)	173 (49.7)	100 (55.9)	ns ^2^
Low risk, *n* (%)	20 (11.6)	15 (15.2)	-
Intermediate favorable risk, *n* (%)	53 (30.6)	32 (32.3)	-
Intermediate unfavorable risk, *n* (%)	47 (27.2)	30 (30.3)	-
High risk, *n* (%)	53 (30.6)	23 (23.2)	-
Clinical significant PCa, *n* (%)	153 (44)	85 (47.5)	ns ^2^

^1^ Number of evaluable samples based on a serum PSA < 2.5 ng/mL. ^2^ Not significant.

**Table 2 cancers-13-03570-t002:** Urinary PSA expression among healthy subjects and patients.

Diagnosis	Mean (ng/mL)	Median (p25–p75)	*p*	*p*
no PCa	4809	1158 (345–2918)	ref	-
Low Risk	2932	2774 (742–4755)	ns ^3^	ref
Int-fav Risk ^1^	1207	338 (150–1112)	0.0111	0.0119
Int-sfav Risk ^2^	1164	371 (41–1297)	0.0018	0.0035
High risk	623	347 (103–704)	0.0001	0.0003
*p* for trend = 0.0027				

^1^ Intermediate favorable risk. ^2^ Intermediate unfavorable risk. ^3^ Not significant.

**Table 3 cancers-13-03570-t003:** Diagnostic models.

Model	AUC ^1^	SE ^2^	95% CI ^3^	*p*	Optimism	Specif ^4^	Specif ^5^
SOC	0.612	0.0309	0.559–0.664	ref	0.001	7.5	11.3
uPSA	0.691	0.0281	0.639–0.739	0.0456	0.001	19	32.8
uPSA + SOC	0.721	0.0271	0.671–0.767	0.0001	0.001	23.4	31.8

^1^ Area under the curve, ^2^ standard error, ^3^ confidence interval, ^4^ specificity at 95% sensitivity; ^5^ specificity at 90% sensitivity.

**Table 4 cancers-13-03570-t004:** Diagnostic accuracy.

Model	95% Sensitivity			90% Sensitivity		
	PPV ^1^	NPV ^2^	NNP ^3^	PPV	NPV	NNP
SOC ^4^	44.7	65.7	9.7	44.4	58.9	30.3
uPSA	47.9	82.8	3.2	51.3	80.7	3.1
uPSA + SOC	49.4	85.7	2.8	50.9	80.2	3.2

^1^ Positive predictive value, ^2^ negative predictive value, ^3^ number need to predict, ^4^ standard of care (sPSA, DRE, age).

**Table 5 cancers-13-03570-t005:** Performance of uPSA in serum PSA subgroups.

Serum PSA Group	Non Cancer (*n*)	csPCa (*n*)	AUC
0–4 ng/mL	32	19	0.602
4.1–10 ng/mL	135	103	0.683
10.1–25 ng/mL	28	31	0.792

**Table 6 cancers-13-03570-t006:** Validation of diagnostic models.

Model	AUC	SE	95% CI	*p*	Specif ^1^	Specif ^2^
SOC	0.598	0.0436	0.427–0.576	ref	8.5	14
uPSA	0.720	0.0373	0.648–0.784	0.025	27.7	29.8
uPSA + SOC	0.666	0.0400	0.592–0.735	0.002	23.4	28.5

^1^ Specificity at 95% sensitivity; ^2^ specificity at 90% sensitivity.

**Table 7 cancers-13-03570-t007:** Validation of diagnostic accuracy.

Model	95% Sensitivity			90% Sensitivity		
	PPV	NPV	NNP	PPV	NPV	NNP
SOC	48.4	65.3	7.3	48.6	60.7	10.8
uPSA	54.3	86.0	2.5	53.7	76.7	3.3
uPSA + SOC	52.9	83.8	2.7	53.2	75.9	3.4

**Table 8 cancers-13-03570-t008:** Clinical performance of urinary PSA.

Cut-Off (probab)	All *n* (%)	No PCa *n* (%)	Low Risk *n* (%)	csPCa *n* (%)	Missed High Risk *n* (%)	Saved Biopsies *n* (%)
0	179 (100)	79 (100)	15 (100)	85 (100)	0 (0)	0 (0)
0.25	163 (91)	63 (80)	15 (100)	85 (100)	0 (0)	16 (17)
0.40	152 (85)	56 (71)	13 (87)	83 (98)	0 (0)	25 (27)
0.50	133 (74)	49 (62)	11 (73)	73 (86)	3 (13)	34 (36)
0.55	85 (45)	21 (27)	7 (47)	57 (67)	7 (30)	66 (70)

**Table 9 cancers-13-03570-t009:** Diagnostic models.

Model	AUC	SE	95% CI	*p*	*p*	*p*
SOC	0.594	0.0419	0.519–0.665	ref	-	-
MRI	0.598	0.0330	0.524–0.670	ns ^1^	ref	-
SOC+MRI	0.629	0.0409	0.555–0.699	ns ^1^	ns ^1^	ref
uPSA	0.695	0.0387	0.623–0.760	0.07	0.05	ns ^1^
uPSA + SOC + MRI	0.698	0.0384	0.626–0.764	0.0055	0.0023	0.0085

^1^ Not significant.

**Table 10 cancers-13-03570-t010:** Clinical performance of urinary PSA in repeat biopsy.

Cut-Off (Probab)	All *n* (%)	No PCa *n* (%)	Low Risk *n* (%)	csPCa *n* (%)	Missed High Risk *n* (%)	Saved Biopsies *n* (%)
0	56 (100)	27 (100)	8 (100)	21 (100)	0 (0)	0 (0)
0.25	48 (86)	22 (81)	6 (75)	20 (95)	0 (0)	7 (20)
0.40	40 (71)	15 (56)	5 (63)	20 (95)	0 (0)	15 (43)
0.50	27 (48)	8 (30)	2 (25)	17 (81)	0 (0)	25 (71)
0.55	0 (0)	0 (0)	0 (0)	0 (0)	4 (100)	35 (100)

## Data Availability

The data presented in this study are available on request from the corresponding author. The data are not publicly available due to privacy restrictions as it contains personal health information of research participants.

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
