# Peer review of "Low Levels of Urinary PSA Better Identify Prostate Cancer Patients"

_cancers, 2021, doi:10.3390/cancers13143570_

Round 1

Reviewer 1 Report

The manuscript describes the new parameter for evaluating prostate cancer levels of urinary PSA. Traditionally the measurement of serum PSA in combination with other clinical parameters is a gold standard for prostate cancer diagnosis and prediction. However, the limitations of the use of these measurements are known.  The urinary PSA level, indeed, better correlates with PSA expression in the prostate tissue and, therefore, can be considered as a better diagnostic prognostic marker. 

In this work, the authors showed that uPSA measurement better than sPSA reflects PC status. This is a really important matter. They used strong statistic analysis for an adequate number of subjects. As they mentioned in the conclusion section, the main limitation is the use of the conventional TRUS-guided biopsy but not MRI/TRUS. The last one is becoming widely useable and, indeed, would be nice to perform a study using this approach. 

The introduction is too long and not focused enough. It is not clear how many pathological samples were investigated. Conclusion are adequate.

Author Response

Thanks for your kind comments. We added in the introduction and discussion sections a paragraph about several prostate cancer biomarkers already investigated and their limitations.

The number of pathological and non-cancer patients were better specified in Section 3.1.1.

Reviewer 2 Report

The authors have prospectively collected patients' urine samples to analyze the PSa level in the urine. The results showed that the PSA in the urine was not enough to differentiate the prostate cancer detection, but gave some infomration about the aggressiveness of the tumor once the patient was diagnosed prostae cancer. To support their results and hypotheses, some additional information would be needed.

Firstly, in the methods section, the authors should describe more detailed information about the handling of the urine samples such as storage temperature and condition, the volume.

Secondly, all the patients received the prostae biopsies. What were the dianoses for the non-cancer patients? how many patients were prostatitis and how many patients received antibiotics before the prostate biopsy?

Thirdly, add the volume of prostate e cancer and some radiologic information about the prostate whether any correlations were observed with the urinary PSA levels

Forthly, Did the authors have any sequential urine samples collected to analyze after fter the prostate biopsy for the non-cancer patients

Lastly, The psa level seems  non-prostae cancer specific analytic exam. Are there any other prostae cancer specifc biomarker for the urinary samples?

Author Response

1. Firstly, in the methods section, the authors should describe more detailed information about the handling of the urine samples such as storage temperature and condition, the volume.

REPLY

In Section 2.2 we reported more detailed procedure about samples collection and storage:

“First 30 ml of voided urine were collected after prostate massage to extract prostatic secretions, through three digital compressions in each lobe starting from the base, moving downwards to the middle and the apex in a timelapse of 30 seconds.

After a gentle shake of the sample an aliquot of 15 ml was taken and stored in Falcon tubes at -80 °C within 5 minutes from collection.”

2. Secondly, all the patients received the prostae biopsies. What were the dianoses for the non-cancer patients? how many patients were prostatitis and how many patients received antibiotics before the prostate biopsy?

REPLY

The diagnoses for non-cancer patients were specified for each cohorts in Section 3.1.1.

Briefly, considering all patients with negative biopsy about 50% were free from disease or with benign hyperplesia, 28% had inflammation/prostatitis, 13% had HGPIN, 9% ASAP.

Of men with already known inflammation/prostatitis about 17% received antibiotics before the prostate biopsy.

3. Thirdly, add the volume of prostate e cancer and some radiologic information about the prostate whether any correlations were observed with the urinary PSA levels

REPLY

No correlation were observed between urinary PSA and prostate volume, both in healthy individuals and cancer patients.

We assessed the possibility to combine urinary PSA analysis with MRI results and conventional routine parameters. In Section 3.1.8 we showed that urinary PSA improves the diagnostic performance of MRI and standard parameters.

As mentioned in discussion, the main limitation of this study is the fact that only a minority of patients was diagnosed with a MRI/TRUS fusion biopsy. Future studies will be designed to specifically test the diagnostic performance of urinary PSA combined with radiologic information. 

4. Forthly, Did the authors have any sequential urine samples collected to analyze after fter the prostate biopsy for the non-cancer patients

REPLY

The study presented here was designed to test the diagnostic capability of urinary PSA in predicting prostate biopsy outcome therefore further urine collections was not expected.

In another study still ongoing we planned to collect sequential urine samples to evaluate the diagnostic capability of urinary PSA and other biomarkers in predicting the outcome of repeated prostate biopsy in non-cancer patients.

Results will be published at the end of the study.

5. Lastly, The psa level seems  non-prostate cancer specific analytic exam. Are there any other prostae cancer specifc biomarker for the urinary samples?

REPLY

PSA is a molecule normally produced by prostate cells. Here we observed that loss of PSA expression and production could represent an hallmark of neoplastic transformation in prostate cells.

In the last few years several prostate cancer biomarkers were identified and studied to ameliorate the management of diagnostic and prognostic path of prostate cancer.

One example is Prostate Cancer Antigen 3 (PCA3), a biomarker that has been clinically validated for the diagnosis of prostate cancer. It is a long non-coding RNA overexpressed by several prostate tumor and detectable in urine.

However, due to high prevalence of high grade tumors with low PCA3 expression in the biopsy setting, its use is limited in few selected cases.

We mention now in the discussion several biomarkers that showed good predictive values in the diagnosis of prostate cancer, but none of that are recommended by guidelines.

To date, there are no reliable urinary biomarkers for prostate cancer.

Reviewer 3 Report

Interesting study

The authors should note that beyond MRI there are multiple ancillary tests with better performance than PSA readily available for decision making regarding biopsy.  These include but are not limited to %free PSA, prostate health index, urinary PCA3, SelectDX, Opko 4K.  All of these tests provide for much better accuracy than PSA and avoidance of unnecessary prostate biopsies.  The authors should mention these tests in the introduction and in the discussion state as a limitation that none of these tests were routinely used in the population.

Author Response

We thank this reviewer for his kind comments. We added in the introduction and discussion sections a paragraph about several prostate cancer biomarkers already investigated and their limitations.

Round 2

Reviewer 2 Report

How were the non-cancer patients with high PSA after biopsy treat ? Please, explain the treatment and follow-up protocol after negative biopsies. Did the patients receive another repat biopsies during the follow-up, and then how many patients diagnosed prostate cancer in the repeat biopsies and what were their urinary PSA levels at the first biopsies?

Try to stratify patients according to the PSA level (4-10 and 10-20 and 20<) and to analyze the relations between serum PSA level and urinary PSa levels? In which categorical groups did the urinary PSA had higher diagnostic efficacy ?

Was the urinary PSa influenced by the body composition such as BMI? Try to stratify the patients’ urinary PSA level into the BMI subgroups.

The urine sample is influence by many factors such as colleting time and fluid intake. How did the authors control the fluid intake and sample collection time such as first morning sample. Did the urinary PSA differ according to the time of collection of urinary sample?

Author Response

  1. How were the non-cancer patients with high PSA after biopsy treat ? Please, explain the treatment and follow-up protocol after negative biopsies. Did the patients receive another repat biopsies during the follow-up, and then how many patients diagnosed prostate cancer in the repeat biopsies and what were their urinary PSA levels at the first biopsies?

Follow-up of patients with previous negative biopsies included dosage of PSA and clinical visits according to physician's preference. Rebiopsy is recommended in presence of a rapid rise in serum PSA and at some point after diagnosis of ASAP.

Some patients with a previous negative biopsy underwent to a second biopsy the our urology clinic.

Concerning the role of uPSA in predicting outcome of a repeat biopsy, additional results were added in section 3.1.9 and Table 10.

Briefly, uPSA displayed a good capability in predicting the outcome of a second biopsy in men with a previous negative prostate biopsy.

  1. Try to stratify patients according to the PSA level (4-10 and 10-20 and 20<) and to analyze the relations between serum PSA level and urinary PSa levels? In which categorical groups did the urinary PSA had higher diagnostic efficacy ?

New results were added in section 3.1.6 and resumed in Table 5.

Briefly, men in Cohort A were divided into three groups, 51 with serum PSA level ≤4 ng/mL, 238 with serum PSA level >4.1 ng/ml and <10 ng/mL, 59 with serum PSA level >10.1 ng/ml and <25 ng/mL. The urinary PSA levels were compared between men with Non-cancer  and patients with csPCa. ROC analysis for uPSA in men with sPSA between 0-4 ng/ml, 4.1-10 ng/ml and 10.1-25 ng/ml were conducted. The AUCs were 0.602, 0.683 and 0.792, respectively.

No correlation between urinary and serum PSA were observed in each group.

  1. Was the urinary PSa influenced by the body composition such as BMI? Try to stratify the patients’ urinary PSA level into the BMI subgroups.

Thank you for your suggestion. Unfortunately we don’t have information about of BMI. We will definitely keep it in consideration for future studies.

  1. The urine sample is influence by many factors such as colleting time and fluid intake. How did the authors control the fluid intake and sample collection time such as first morning sample. Did the urinary PSA differ according to the time of collection of urinary sample?

Urine samples were collected in the morning during the urological examination in which the prostate biopsy was performed.

Urine were collected after a prostate massage by which prostate secretions were pushed directly into the urethra and flushed out of the body on urination.

Based on the literature the recovery of prostate contents in urine is dependent on prostate manipulation. We have observed that in the absence of prostate massage, PSA is detectable at low levels, both in patients and in healthy donors.

PSA concentration was evaluated in first catch urine immediately after prostate massage and fixed volume of urine (30 ml) were collected. For this reason the amount of uPSA reported was not dependent on fluid intake.